# Integrating Distributed Acoustic Sensing and PINN Frameworks for Enhanced Indoor Sound Source Localization

## Abstract

Distributed Acoustic Sensing (DAS) is an emerging technology that transforms standard optical fibers into dense arrays of acoustic sensors, offering unprecedented opportunities for smart city applications, indoor monitoring of human activity, and surveillance without compromising privacy. In this paper, we integrate DAS with Physics-Informed Neural Networks (PINNs) for indoor sound source localization. By embedding the acoustic wave equation and impedance boundary conditions into the neural network architecture, we exploit physical laws to guide the learning process, improving accuracy and generalization. We propose two strategies for real-time sound source localization using DAS data. The first strategy involves training the PINN on all available data simultaneously, while the second strategy incrementally feeds data over time, simulating real-time data acquisition. Using real indoor DAS measurements, we demonstrate the effectiveness of our approach in deciphering complex room acoustics and accurately inferring sound source locations under both strategies. Our framework provides a novel solution for real-time indoor positioning and human activity surveillance, offering significant advantages over traditional camera-based systems by preserving individual privacy.

## 1 Introduction

Distributed Acoustic Sensing (DAS) is an emerging technology that transforms standard optical fibers into dense arrays of acoustic sensors, offering unprecedented opportunities for environmental monitoring, urban infrastructure analysis, and indoor activity surveillance in smart cities (Lindsey & Martin, 2021; Li et al., 2022). By capturing acoustic signals along the length of optical fibers, DAS enables the detection and localization of acoustic events without the need for traditional sensor networks, providing a cost-effective and privacy-preserving solution for monitoring applications. By "privacy-preserving," we mean that optical fibers can detect human activities, such as movement trajectories, without being able to intelligently interpret human speech due to the limited frequency band. Beyond urban environments, DAS technology has demonstrated its versatility in various fields, including ultra-long-distance power cable monitoring without inline amplification (Cedilnik & Lees, 2019), volcanic event detection and imaging of near-surface volcanic structures (Jousset et al., 2020), and underwater seismic research transforming telecommunication cables into seismo-acoustic sensors (Lior et al., 2021). These advancements highlight the potential of DAS in improving infrastructure monitoring, hazard assessment, and contributing to earthquake early warning systems. In this work, we focus on leveraging DAS for indoor activity surveillance, aiming to detect and localize human movements within smart city environments. By integrating DAS with a physics-informed learning framework, we reduce the dependence on labeled data and enhance the generalization capabilities of our model, providing valuable insights while preserving privacy.

Physics-Informed Neural Networks (PINNs) have gained attention as a powerful computational framework for solving forward and inverse problems governed by partial differential equations (PDEs) (Raissi et al., 2019). By embedding physical laws directly into the neural network training process, PINNs leverage both data and governing equations to model complex phenomena without requiring large labeled datasets or computational meshes. This approach has been successfully applied in various fields, including fluid dynamics, heat transfer, and wave propagation. Subsequent

research has introduced various modifications to PINNs to enhance their accuracy, efficiency, and scalability (Lu et al., 2021; Cuomo et al., 2022). Variational hp-VPINN extends the PINN framework by incorporating domain decomposition methods, allowing for more efficient and accurate solutions, especially for problems with complex geometries or high-dimensional PDEs (Kharazmi et al., 2021). Conservative PINN (CPINN) focuses on preserving the conservation laws inherent in certain PDEs, ensuring that the learned solutions respect these fundamental physical principles, which is crucial for applications involving conservation of mass, momentum, or energy (Jagtap et al., 2020). Each of these variants aims to enhance the versatility and robustness of PINNs, making them more suitable for a wider range of scientific computing problems. There are a couple of drawbacks for many PINNs with soft constraints for boundary conditions (BCs)BCs and initial conditions (ICs)ICs. The selection of weights and samples for BCs and ICs cannot certainly be determined and requires many trial-and-error tests. Even when the loss function is minimized, the BCs and ICs are not strictly satisfied. A recent analysis of PINNs has revealed potential failure modes, particularly when dealing with complex problems involving convection, reaction, and diffusion operators (Krishnapriyan et al., 2021). To target the scaling problems of general PDEs and take advantage of parallel computing, XPINNs and FBPINNs have been developed based on domain decomposition methods (Jagtap & Karniadakis, 2020; Shukla et al., 2021; Moseley et al., 2023).

In the context of acoustics, PINNs offer the potential to model sound wave propagation and solve inverse problems such as source localization. Previous studies have explored the use of PINNs for solving the Helmholtz equation and other wave equations, demonstrating the capability of PINNs to capture wave phenomena (Jagtap & Karniadakis, 2020; Shukla et al., 2021; Moseley et al., 2023). However, challenges persist in applying PINNs to real-world acoustic problems, including:

- Sensitivity to Hyperparameters: The performance of PINNs is highly sensitive to the choice of hyperparameters, such as the weights assigned to different loss terms, which can affect convergence and accuracy.
- Difficulty in Enforcing Boundary and Initial Conditions: When boundary and initial conditions are imposed as soft constraints in the loss function, PINNs may struggle to satisfy these conditions accurately, leading to suboptimal solutions.
- Scalability Issues: Scaling PINNs to complex or large-scale domains remains challenging due to computational constraints and optimization difficulties.

To address these challenges, various modifications to the PINN framework have been proposed. Extended PINNs (XPINNs) and Finite Basis PINNs (FBPINNs) utilize domain decomposition and parallel computing to enhance scalability and efficiency (Jagtap & Karniadakis, 2020; Shukla et al., 2021; Moseley et al., 2023). These approaches aim to improve training performance and enable the application of PINNs to more complex problems.

**Contribution** In this paper, we integrate DAS technology with PINNs to develop a framework for indoor sound source localization and spatiotemporal wave field recovery. Our main contributions are:

- We propose a PINN framework tailored for modeling room acoustics, incorporating the acoustic wave equation and impedance boundary conditions to guide the learning process.
- We demonstrate the simultaneous localization of sound sources and recovery of the spatiotemporal acoustic field using real DAS measurements, highlighting the effectiveness of our approach in complex indoor environments.
- We explore the effects of different training strategies—including incremental data feeding—and fiber array configurations on localization accuracy, providing insights into optimal system design and deployment.

**Related Work** Recent advancements have seen the application of PINNs to various wave-related phenomena. For instance, Moseley et al. (2023) employed PINNs for solving wave equations in heterogeneous media, while Mao et al. (2020) addressed high-speed flows and shock wave modeling using PINNs. In acoustics, Olivieri et al. (2024) utilized PINNs for sound field reconstruction but did not consider impedance boundary conditions. Furthermore, the work by Borrel-Jensen et al. (2024) presents a novel approach to sound propagation in realistic interactive 3D scenes using deep neural operators, which is particularly relevant to the challenges faced in virtual acoustics.

In terms of source localization, traditional methods often rely on time-of-arrival or beamforming techniques, which require extensive sensor networks and can be computationally intensive (Dibiase, 2000). Zhai et al. (2023) proposed an innovative approach to enhance localization accuracy in reverberant environments by developing a grid-free global optimization algorithm that combines the image source model and state transition algorithm.

Our work differentiates itself by integrating DAS with a physics-informed learning framework, reducing the dependence on labeled data and leveraging the underlying physics for improved generalization. By addressing the challenges associated with PINNs in real-world applications, we advance the state-of-the-art in acoustic source localization and provide a foundation for future research in this area.

## 2 DATA FROM DISTRIBUTED ACOUSTIC SENSING

Distributed Acoustic Sensing (DAS) leverages the sensitivity of optical fibers to external perturbations to detect acoustic signals along the length of the fiber. When a laser pulse is sent through an optical fiber, imperfections and variations within the fiber cause backscattering of light through Rayleigh scattering. External acoustic vibrations induce strain in the fiber, leading to changes in the backscattered light intensity, phase, or polarization, which can be detected and related to the external disturbances (Hartog, 2017).

In this study, we strategically deployed a 2-kilometer-long optical fiber network within a five-story building located on the campus of Tsinghua University (Figure 1). The network spans each level of the structure, ranging from the roof, at an elevation of $17.45\,\text{m}$, down to the basement at $-3.5\,\text{m}$ below ground level. A segment of the fiber optic cable, extending $400\,\text{m}$ in length, was buried at a depth of approximately $0.5\,\text{m}$ in the building's adjacent yard. This strategic placement facilitates comprehensive four-dimensional (4D) spatio-temporal monitoring of the building's structural integrity and dynamic responses.

To achieve high-resolution spatio-temporal monitoring, the DAS system was configured with a gauge length of $3.2\,\text{m}$ and a spatial resolution of $1.6\,\text{m}$, effectively creating a network of $1,250$ distinct monitoring channels. This setup ensures high-fidelity capture of structural dynamics, with a data acquisition rate of $1,000\,\text{Hz}$. The precise positioning of each channel was determined through a tap test experiment, further validated against the architectural floor plans of the building.

The fiber layout and channel distribution are depicted in Figure 1, illustrating the extensive coverage achieved within the building. The developed PINN framework, integrating the governing acoustic wave equations with the DAS strain measurements, provides a powerful tool for interpreting real DAS data. By embedding physical laws into the neural network, we reduce the dependence on extensive labeled datasets and enhance the generalization capabilities of the model.

Developing a PINN framework to decipher DAS data collected from the building, we can reconstruct the spatio-temporal acoustic pressure fields and accurately localize sound sources within the structure. This capability is crucial for monitoring human activities, detecting structural anomalies, and enhancing surveillance without compromising privacy.

## 3 GOVERNING EQUATIONS AND TRAINING STRATEGIES

The propagation of acoustic waves in a three-dimensional domain $\Omega \subset \mathbb{R}^3$ with boundary $\partial\Omega$ is governed by the acoustic wave equation:

$$\nabla^2 p(\mathbf{x}, t) - \frac{1}{c^2} \frac{\partial^2 p(\mathbf{x}, t)}{\partial t^2} = 0, \quad \mathbf{x} \in \Omega, \ t \in [0, T], \tag{1}$$

where $p(\mathbf{x}, t)$ is the acoustic pressure at position $\mathbf{x} = (x, y, z)$ and time $t$, and $c$ is the speed of sound in the medium.

To fully describe the problem, we impose the following initial conditions (ICs):

$$p(\mathbf{x}, 0) = I_1(\mathbf{x}), \quad \mathbf{x} \in \Omega, \tag{2}$$

$$\frac{\partial p}{\partial t}(\mathbf{x}, 0) = I_2(\mathbf{x}), \quad \mathbf{x} \in \Omega, \tag{3}$$

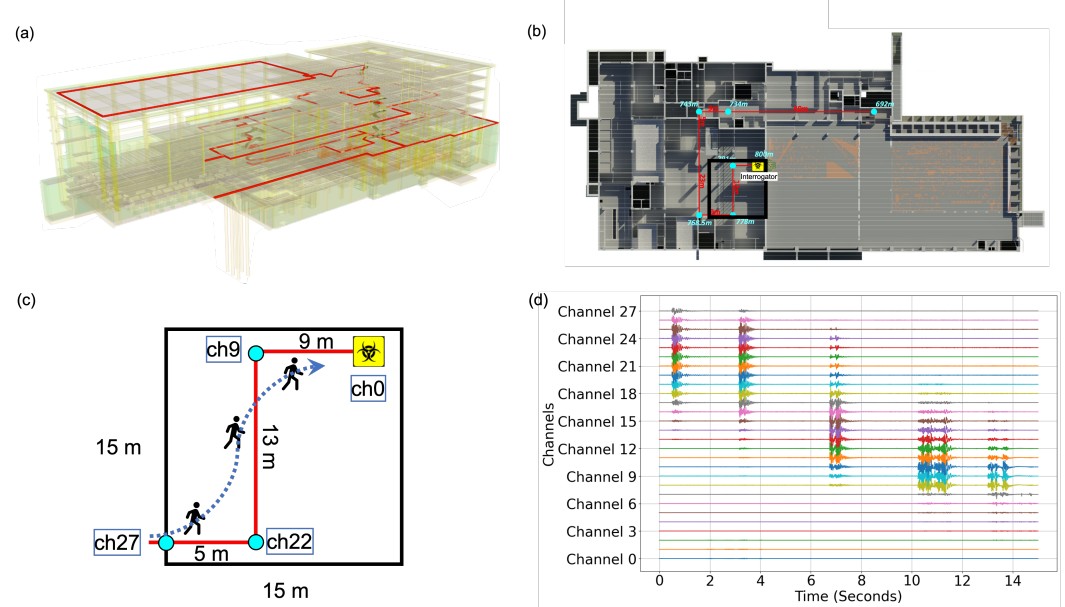

Figure 1: (a) 3D model illustrating the layout of fiber optic arrays across multiple floors. (b) A 100-meter-long fiber optic array on a single floor. The black square denotes the room we detected human walking trajectory. (c) A person entering a room of size 15 m by 15 m, tracked through the fiber optic sensing system. The fibers in this squared room is shown in red lines. Locations of Channle 0, 9, 22, 27 are marked with blue circles. (d) Recorded acoustic waves induced by human footprints. The signal appears sequentially from channel 27 (near the door) to channel 7 (close to the DAS interrogator), reflecting multiple sources of a moving target.

where $I_1(\mathbf{x})$ and $I_2(\mathbf{x})$ represent the initial pressure distribution and its time derivative, respectively. Specifically, we model the initial pressure distribution as a Gaussian function centered at the unknown source location $\mathbf{x}_0 = (x_0, y_0, z_0)$:

$$I_1(\mathbf{x}) = A \exp\left(-\frac{\|\mathbf{x} - \mathbf{x}_0\|_2^2}{\sigma^2}\right), \tag{4}$$

where $A$ is the amplitude and $\sigma$ controls the width of the Gaussian.

The boundary conditions (BCs) are given by the impedance boundary condition:

$$Z\frac{\partial p}{\partial n}(\mathbf{x}, t) + \rho_0 \frac{\partial p}{\partial t}(\mathbf{x}, t) = 0, \quad \mathbf{x} \in \partial\Omega, \ t \in [0, T], \tag{5}$$

where $Z$ is the acoustic impedance of the boundary, $\rho_0$ is the ambient density, and $\partial p/\partial n$ denotes the normal derivative at the boundary.

**Physics-Informed Neural Network Framework**   We employ a Physics-Informed Neural Network (PINN) to approximate the acoustic pressure $p(\mathbf{x}, t)$ and to infer the unknown source location $\mathbf{x}_0$. The neural network takes spatial coordinates and time as inputs and outputs the predicted pressure:

$$p_\theta(\mathbf{x}, t) = \mathrm{NN}_\theta(\mathbf{x}, t), \tag{6}$$

where $\theta$ represents the trainable parameters of the neural network.

To enforce the governing equations, initial conditions, and boundary conditions, we define the following residuals:

The total loss function $\mathcal{L}(\theta)$ combines the residuals using mean squared errors:

$$\mathcal{L} = \lambda_{\text{pde}} \cdot \mathcal{L}_{pde} + \lambda_{\text{ic}} \cdot \mathcal{L}_{ic} + \lambda_{\text{bc}} \cdot \mathcal{L}_{bc} + \lambda_{\text{data}} \cdot \mathcal{L}_{data}$$

$$= \lambda_{\text{pde}} \frac{1}{N_{pde}} \sum_{i=1}^{N_{pde}} \left\| \nabla^2 p(\mathbf{x}_{pde}^i, t_{pde}^i) - \frac{1}{c^2} \frac{\partial^2 p(\mathbf{x}_{pde}^i, t_{pde}^i)}{\partial t^2} \right\|_2^2$$

$$+ \lambda_{\text{bc}} \frac{1}{N_{bc}} \sum_{i=1}^{N_{bc}} \left\| Z \frac{\partial p}{\partial n}(\mathbf{x}_{bc}^i, t_{bc}^i) + \rho_0 \frac{\partial p}{\partial t}(\mathbf{x}_{bc}^i, t_{bc}^i) \right\|_2^2 \qquad (7)$$

$$+ \lambda_{\text{ic1}} \frac{1}{N_{ic1}} \sum_{i=1}^{N_{ic1}} \left\| p(\mathbf{x}_{ic}^i, 0) - I_1(\mathbf{x}_{ic}^i; \mathbf{x}_0) \right\|_2^2 + \lambda_{\text{ic2}} \frac{1}{N_{ic2}} \sum_{i=1}^{N_{ic2}} \left\| \frac{\partial p(\mathbf{x}_{ic}^i, 0)}{\partial t} - I_2(\mathbf{x}_{ic}^i; \mathbf{x}_0) \right\|_2^2$$

$$+ \lambda_{\text{data}} \frac{1}{N_{data}} \sum_{i=1}^{N_{data}} \left\| \dot{\epsilon}_{\xi\xi}(\mathbf{x}_{data}^i, t_{data}^i) + \frac{2\nu p(\mathbf{x}_{data}^i, t_{data}^i)}{E} \right\|_2^2,$$

where $\lambda_{\text{pde}}$, $\lambda_{\text{ic}}$, $\lambda_{\text{bc}}$, and $\lambda_{\text{data}}$ are positive weighting coefficients that balance the contributions of each loss component during training. The terms $\mathcal{L}_{\text{pde}}$, $\mathcal{L}_{\text{ic}}$, $\mathcal{L}_{\text{bc}}$, and $\mathcal{L}_{\text{data}}$ represent the losses associated with the governing partial differential equation (PDE), initial conditions (ICs), boundary conditions (BCs), and observational data, respectively. ~~$\lambda_{\text{pde}}$, $\lambda_{\text{ic1}}$, $\lambda_{\text{ic2}}$, $\lambda_{\text{bc}}$, and $\lambda_{\text{data}}$ are weighting coefficients, and $p_{\text{data}}(\mathbf{x}, t)$ represents the observed pressure data from DAS measurements.~~ The first term, $\mathcal{L}_{\text{pde}}$, enforces the acoustic wave equation at selected collocation points. Specifically, it ensures that the predicted pressure field $p(\mathbf{x}, t)$ satisfies the PDE by minimizing the residual at $N_{\text{pde}}$ collocation points $(\mathbf{x}_{\text{pde}}^i, t_{\text{pde}}^i)$. The second term, $\mathcal{L}_{\text{bc}}$, enforces the boundary conditions (Equation 5) at $N_{\text{bc}}$ boundary collocation points $(\mathbf{x}_{\text{bc}}^i, t_{\text{bc}}^i)$. The third term comprises two components, $\mathcal{L}_{\text{ic1}}$ and $\mathcal{L}_{\text{ic2}}$, which enforce the initial conditions at $t = 0$ across $N_{\text{ic1}}$ and $N_{\text{ic2}}$ collocation points $(\mathbf{x}_{\text{ic}}^i, 0)$, respectively. The fourth term, $\mathcal{L}_{\text{data}}$, incorporates the observational data from the Distributed Acoustic Sensing (DAS) system into the training process. It ensures that the predicted pressure field is consistent with the measured strain rate data $\dot{\epsilon}_{\xi\xi}$ at $N_{\text{data}}$ data points $(\mathbf{x}_{\text{data}}^i, t_{\text{data}}^i)$. The geometry of the DAS array is described by the equation $\xi(\mathbf{x}) = 0$. The strain rate along the axial direction of the fiber is

$$\dot{\epsilon}_{\xi\xi} = \frac{-2\nu p(\mathbf{x}, t)}{E}, \qquad (8)$$

where $\nu$ is Poisson's ratio, and $E$ is Young's modulus. The architecture of the PINN is shown in Figure 2.

We jointly optimize the neural network parameters $\theta$ and the source location $\mathbf{x}_0$ by minimizing the chosen loss function:

$$\theta^*, \mathbf{x}_0^* = \arg\min_{\theta, \mathbf{x}_0} \mathcal{L}(\theta). \qquad (9)$$

To handle the unknown source location within the PINN framework, we treat $\mathbf{x}_0$ as trainable parameters. During optimization, $\mathbf{x}_0$ is updated alongside $\theta$ using gradient-based methods enabled by automatic differentiation.

**Traning Strategies**  We propose two training strategies to optimize the parameters of the neural network $\theta$ and to infer the location of the source $\mathbf{x}_0$.

**Strategy 1: Simultaneous Training with All Data**  In this strategy, we train the PINN using all available data across the entire time domain simultaneously. The loss function incorporates residuals computed over the full spatial-temporal domain:

$$\mathcal{L}_{\text{Strategy 1}}(\theta) = \mathcal{L}(\theta). \qquad (10)$$

This approach uses all the data at once after completing data streaming corresponding to an interesting acoustic event, potentially leading to significant processing time and computational resources.

**Strategy 2: Incremental Time-Stepping Training**  Strategy 2 simulates real-time data acquisition by incrementally feeding data over time. At each time step $t_k$, we train the PINN using observed

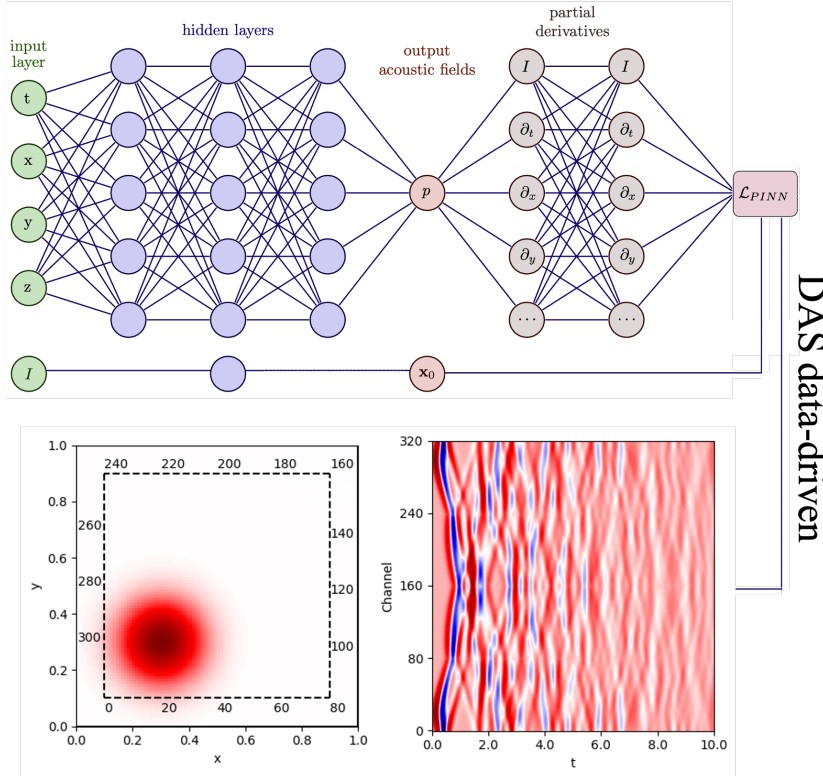

Figure 2: A PINN architecture incorporating DAS observation data.

DAS data up to the current time, updating the loss function accordingly:

$$
\mathcal{L}_{\text{Strategy 2}}^{(k)}(\theta; \mathbf{x}_0) = \lambda_{\text{pde}} \cdot \mathcal{L}_{pde} + \lambda_{\text{ic}} \cdot \mathcal{L}_{ic} + \lambda_{\text{bc}} \cdot \mathcal{L}_{bc} + \lambda_{\text{data}} \cdot \mathcal{L}_{data}^{(k)}
$$

$$
= \lambda_{\text{pde}} \cdot \mathcal{L}_{pde} + \lambda_{\text{ic}} \cdot \mathcal{L}_{ic} + \lambda_{\text{bc}} \cdot \mathcal{L}_{bc} + \lambda_{\text{data}} \cdot \mathbb{E}_{\mathbf{x}, t \leq t_k} \left[ \left( p_{\theta; \mathbf{x}_0}(\mathbf{x}, t) - p_{\text{data}}(\mathbf{x}, t) \right)^2 \right].
$$
(11)

At each incremental step, we optimize $\theta$ using the data available up to $t_k$, refining the model progressively as new data become available. This strategy emulates online learning and can be more computationally efficient per time step.

It is important to note that during the early training epochs of the *Incremental Time-Stepping Training Strategy* (Strategy 2), when we use only the first few time snapshots, the acoustic waves have not yet reached the room boundaries. As a result, the boundary conditions have minimal or no effect on the wave propagation during this initial phase. This effectively simulates an environment without boundary reflections, akin to optimization without boundary conditions. As more time snapshots are added to training progresses, the waves begin to interact with the walls, and the influence of the boundary conditions becomes significant.

## 4 SYNTHETIC TESTS

To validate the proposed PINN framework and its ability to localize sound sources using DAS data, we conduct synthetic experiments in both 2D and 3D environments.

### 4.1 2D SQUARE ROOM

We consider a two-dimensional (2D) square room of size $1 \, \text{m} \times 1 \, \text{m}$. The sound speed $c$ is set to $343 \, \text{m/s}$, and the density $\rho_0$ is $1.2 \, \text{kg/m}^3$. A point source is placed at an unknown location $\mathbf{x}_0 = (0.3, 0.3)$, emitting a Gaussian pulse at $t = 0$ with amplitude $A = 1$ and standard deviation $\sigma = 0.4$.

We use a small neural network with two layers, each containing 64 neurons and utilizing the tanh activation function.

Two DAS array geometries—*Square* and *Halton*—are shown in Figure 3. A total of 320 observation stations are located along the fibers, marked with green triangles in the figure. In the square array, the fiber forms a 2D square shape. In the Halton array, the fiber forms a curve $\xi(x, y)$ such that the spatial locations sampled ideally follow the Halton sequence.

We apply the two training strategies described in the previous section to these synthetic DAS array observations. The inverted source location results are shown in Figure 4. With the same number of DAS channels, we find that a more random spatial coverage results in more accurate and stable localization. These results highlight the importance of the optimal design of DAS array deployment given a fixed number of observation channels.

In comparing the training strategies, the *Incremental Time-Stepping Training Strategy* (Strategy 1) significantly outperforms the *Simultaneous Training with All Data Strategy* (Strategy 2) in achieving stable localization. An interesting observation is that the simultaneous training strategy can lead to final estimated source locations that are symmetric to the true location relative to the domain center. This phenomenon is due to the similarity of acoustic wavefields after multiple wall reflections. When we use observations overall time at once for training, the inverted source location may fall into these symmetric locations. However, the *Simultaneous Training with All Data Strategy* (Strategy 1) performs better for wavefield reconstruction across the entire time domain, capturing wave propagation and reflections more accurately as shown Figure 8 in Appendix A.1. This indicates that Strategy 2 is advantageous for quick source localization with limited data, while Strategy 1 provides a more comprehensive reconstruction of the wavefield when full temporal data is available.

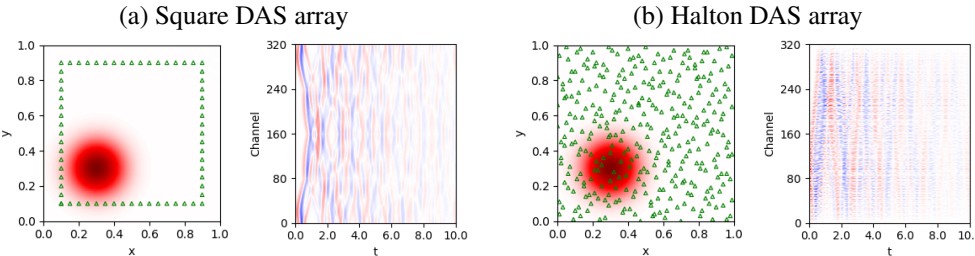

(a) Square DAS array            (b) Halton DAS array

Figure 3: (a) 320 observation stations along the square DAS array with the Finite Element Method (FEM)~~FEM~~ synthetic waveforms denoted by green triangles. (b) 320 observation stations along the Halton DAS array with the FEM synthetic waveforms.

### 4.2 3D CUBE ROOM

To evaluate the performance of the proposed sound source localization framework in three dimensions, we design synthetic tests within a cube room of size $1\,\text{m} \times 1\,\text{m} \times 1\,\text{m}$. The sound speed $c$ is set to $343\,\text{m/s}$, and the density $\rho_0$ is $1.2\,\text{kg/m}^3$. A point source is placed at an unknown location $\mathbf{x}_0 = (0.3, 0.3, 0.3)$, emitting a Gaussian pulse at $t = 0$ with amplitude $A = 1$ and standard deviation $\sigma = 0.4$.

We use a neural network with four layers, each containing 64 neurons and utilizing the tanh activation function.

Three different DAS array geometries—*Edges*, *Floor*, and *Volume*—are illustrated in Figure 6. Each array consists of 512 observation stations located along the fibers, represented by green triangles in the figure. The *Edges* array places fibers along the edges of the walls, the *Floor* array places fibers on the floor of the room, and the *Volume* array distributes fibers throughout the 3D space.

The finite-element modeling of the 3D acoustic wave propagation is shown in Figure 5. The emitted acoustic waves propagate through the environment and are detected by the DAS arrays.

As in the 2D experiment, we apply the two training strategies described earlier to these synthetic DAS array observations. ~~Strategy 2 converges to the correct 3D source location within 2000 epochs.~~

(a) Square DAS array with Strategy 1 (b) Halton DAS array with Strategy 1

(c) Square DAS array with Strategy 2 (d) Halton DAS array with Strategy 2

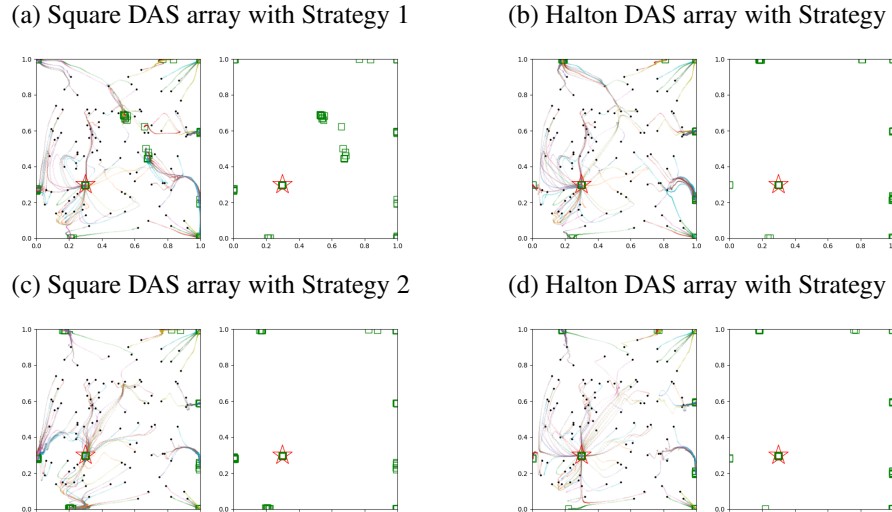

Figure 4: The inverted source locations after 2000 epochs for (a) square DAS array with Strategy 1; (b) Halton DAS array with Strategy 1; (c) square DAS array with Strategy 2; (d) Halton DAS array with Strategy 2. The green squares indicate the locations pinpointed by the location of the sound source under different initial values. The red pentagram represents the actual position of the sound source. It can be observed that DAS-PINN either locates the sound source at its true position or at its conjugate position, both of which result in relatively small PDE-loss and data-loss. The trajectory in the left image represents the change from different initial guessed positions (black dots) to the final converged positions (green squares) during the training process.

~~Just like the 2D test, we observe that the *Incremental Time-Stepping Training Strategy* (Strategy 2) significantly outperforms the *Simultaneous Training with All Data Strategy* (Strategy 1) in achieving stable and accurate localization.~~ For the training process of Strategy 2, we added data every 1000 epochs from one subsequent time snapshot, incrementally providing more temporal information to the model. As shown in Figure 6, the *Edges* and *Floor* cases do not show good localization results with only 2000 epochs. This is because the waves have not propagated to the receivers in these configurations during the early time steps used for training.

In the *Volume* case, with comprehensive data coverage throughout the 3D space, the PINN achieves high accuracy in source localization across all three dimensions. The pressure field reconstruction is highly accurate, demonstrating the effectiveness of the PINN with well-distributed data. The inverted source location results are presented in Figure 6(f).

Consistent with the 2D experiments, Strategy 2 demonstrates superior performance in the 3D localization tasks. Strategy 1 often converges to incorrect locations due to the complexities introduced by simultaneous reflections and wave interactions within the 3D space. Strategy 2, by incrementally incorporating data over time, allows the PINN to more effectively disentangle the wave propagation patterns and accurately infer the source location.

## 5 REAL DATA

We applied our method to real DAS recordings of a person entering a room, as shown in Figure 1 (c) and (d). The human footsteps are treated as a moving target. The room is a 15 m by 15 m square with 28 DAS channels deployed along the floor, which is a subset of the entire cable system of the building. We apply the *Incremental Time-Stepping Training Strategy* (Strategy 2) to locate the moving target, as illustrated in Figure 1. Since the target is moving along the floor, we enforce the constraint $z_0 = 0$ in the inversion.

Using the first ten time snapshots of the DAS recording, and training over 10,000 epochs (about 460 seconds using an A800 GPU), the locations of five events due to a moving target all successfully

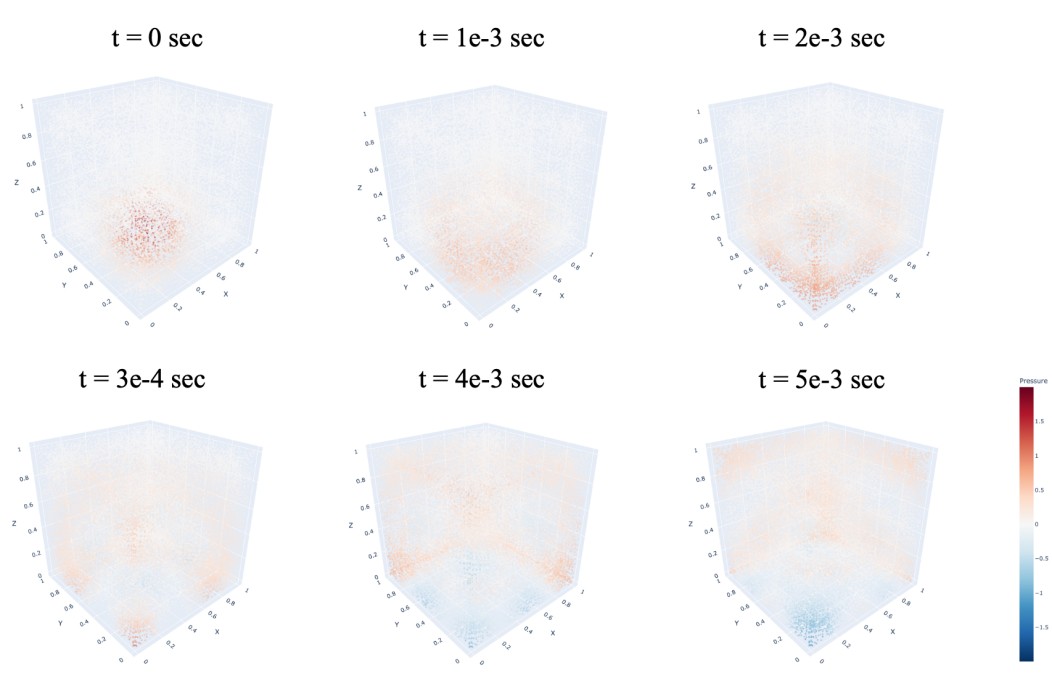

Figure 5: 3D acoustic wave propagation in a cube room. The sound source is located at $x_0 = y_0 = z_0 = 0.3$.

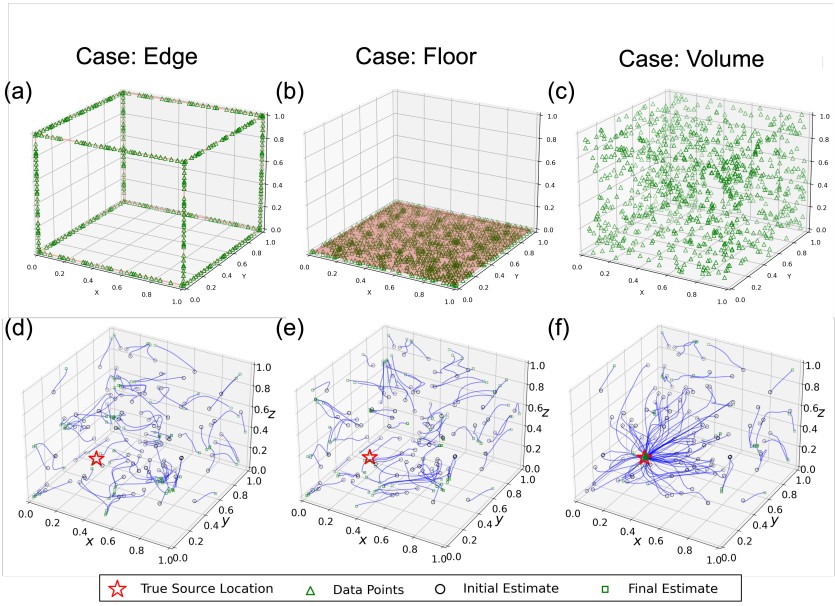

Figure 6: DAS arrays and source localization trajectories for 3D synthetic tests.(a)-(c): Different fiber optic array configurations: (a) Case Edge – fibers are placed along the edges of the room's walls; (b) Case Floor – fibers are deployed on the floor of the room; (c) Case Volume – fibers are distributed throughout the 3D volume of the space. (d)-(f): Localization results for the respective configurations: (d) Case Edge, (e) Case Floor, and (f) Case Volume. The red star marks the true source location, green triangles represent the data points (fiber locations), black circles indicate initial estimates, and green squares represent final estimates after optimization. Blue curves show the trajectories of the estimates converging to the final solution.

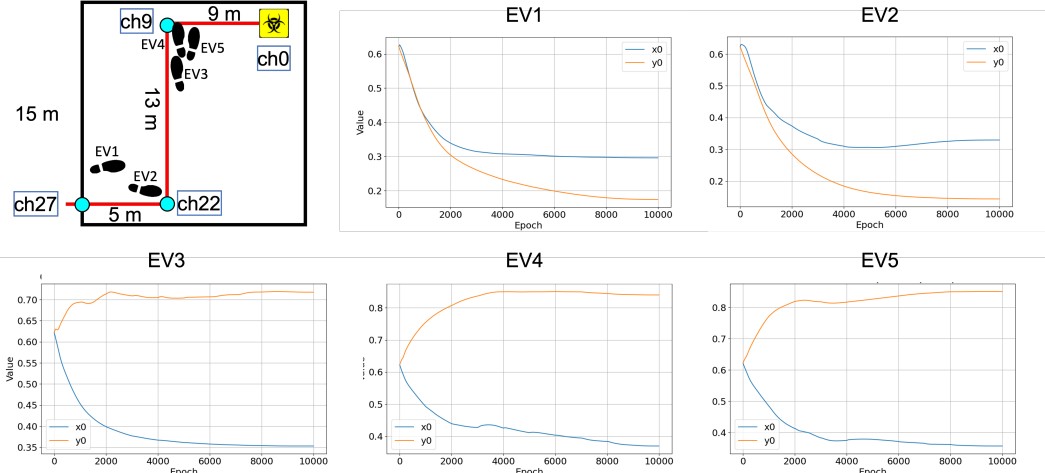

Figure 7: Real DAS data analysis of a moving target. The top-left panel shows the floor plan of the 15 m by 15 m squared room with 28 DAS channels, where the red lines represent the fiber deployment. The moving target (human footsteps) is treated as a source of acoustic signals, with channels labeled from 0 to 27. The remaining panels display the convergence of the normalized $x_0$ and $y_0$ coordinates of the target location over 10,000 training epochs using the *Incremental Time-Stepping Training Strategy (Strategy 2)*. Each plot corresponds to a specific segment of the DAS recording, with the blue line representing $x_0$ and the orange line representing $y_0$.

converged, as shown in Figure 7. Some events converged only with 2000 epochs because of a better initial guess. This means that we can further improve the method by introducing advanced initialization algorithms for the source locations.

## 6   CONCLUSION

The synthetic tests conducted in Sections 4.1 and 4.2 are designed to emulate the conditions of real-world DAS data acquisition in building rooms. By modeling acoustic wave propagation and interaction with boundaries in controlled environments, we establish a foundation for interpreting the complex wavefields captured by the DAS system in the building.

The synthetic tests demonstrate that the PINN framework can effectively localize sound sources using DAS measurements. The accuracy depends on the spatial distribution of data points. Incorporating the source location as trainable parameters allows simultaneous inversion of the source position and pressure field.

Beyond synthetic data, we applied our method to real DAS recordings of a person entering a room, treating human footsteps as a moving target. Using the *Incremental Time-Stepping Training Strategy* (Strategy 2) and enforcing the constraint $z_0 = 0$, we successfully located the moving target within a 15 m by 15 m room equipped with 28 DAS channels. The target's location converged using only a couple of early DAS recordings, demonstrating the practical applicability of our approach to real-world scenarios.

Our synthetic experiments , both synthetic and real,  validate the proposed PINN framework's capability to accurately localize sound sources and reconstruct pressure fields using DAS data. ~~The results highlight the potential of integrating PINNs with DAS technology for advanced indoor sensing applications.~~ This novel framework simultaneously constructs the acoustic wavefield and performs inversion without requiring labeled data. This capability highlights the potential of integrating PINNs with DAS technology for advanced indoor sensing applications, such as real-time tracking of moving targets.

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

# A    APPENDIX

## A.1    RECONSTRUCTED WAVEFIELDS

We compared the reconstructed wavefields using two training strategies. As shown in Figure 8, Strategy 1 (Simultaneous Training with All Data) performs better for wavefield reconstruction. The reconstructed wavefields closely resemble the true wavefields across all time steps, accurately capturing wave propagation and reflections; however, the location is accurate. In contrast, Strategy 2 (Incremental Time-Stepping Training Strategy) performs better for localization using data from early time steps. By focusing on early time snapshots, Strategy 2 enables rapid convergence of the source location parameters. However, the wavefield reconstruction quality is somewhat compromised compared to Strategy 1, especially at later time steps, due to the limited temporal information used during training. When choosing between the two strategies, these observations highlight a trade-off between accurate source localization and comprehensive wavefield reconstruction.

## A.2    CONVERGENCE PLOTS

To further evaluate our PINN framework's performance and the training strategies' effectiveness, we present convergence plots for both synthetic and real data cases. These plots illustrate how the individual loss components and the estimated source locations evolve during the training process.

Figures 9 show the convergence curves for the 2D and 3D synthetic data cases, respectively. We observe that all individual loss components decrease steadily over training epochs. The PDE loss, IC loss, ICP loss, and BC loss decrease gradually, reflecting the model's ability to satisfy the governing physical equations and the boundary and initial conditions. The DAS loss, associated with the data fitting at sensor locations, decreases significantly, demonstrating the model's capability to fit the observed data. The source location estimates converge toward the true source location over epochs, confirming the effectiveness of the PINN framework in accurately localizing the sound source.

Figure 10 shows the convergence curves for the five real DAS data events. For each event, we present the convergence of individual loss components and the estimated source locations.

(a) Reconstructed 2D wavefield for strategy 1

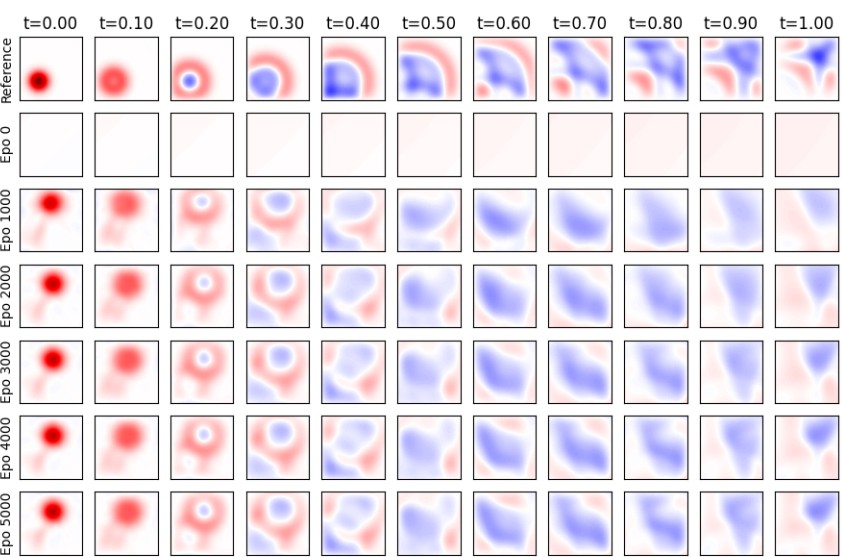

(b) Reconstructed 2D wavefield for strategy 2

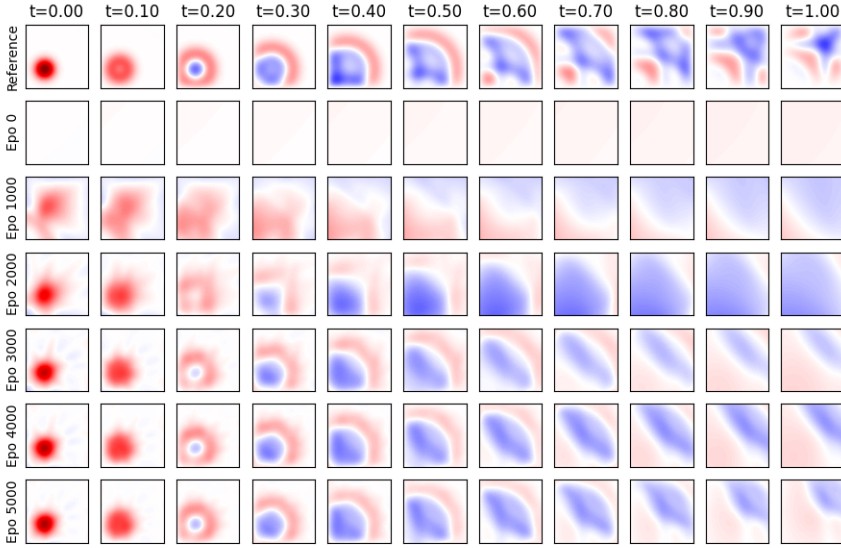

Figure 8: (a) and (b) illustrate the reconstructed 2D wavefields for strategies 1 and 2 at different time steps. The first row in each panel shows the true wavefield for reference. Each subplot corresponds to a specific time step $t = 0.00, 0.10, \ldots, 1.00$. Rows represent the evolution of the wavefield reconstruction across training epochs. The true location is at $\mathbf{x}_0 = (0.3, 0.3)$.

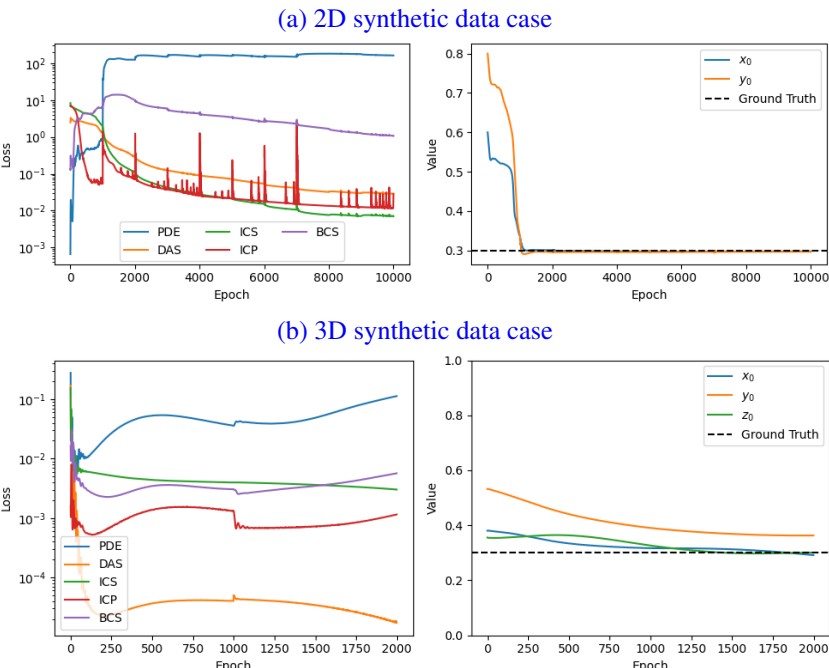

Figure 9: Convergence plots for the (a) 2D and (b) 3D synthetic data case. The left subplot presents the individual loss components convergence over epochs, including PDE loss, IC1(ICS) loss, IC2(ICP) loss, BC loss, and DAS loss, while the right subplot presents the convergence of the estimated source location coordinates over epochs.

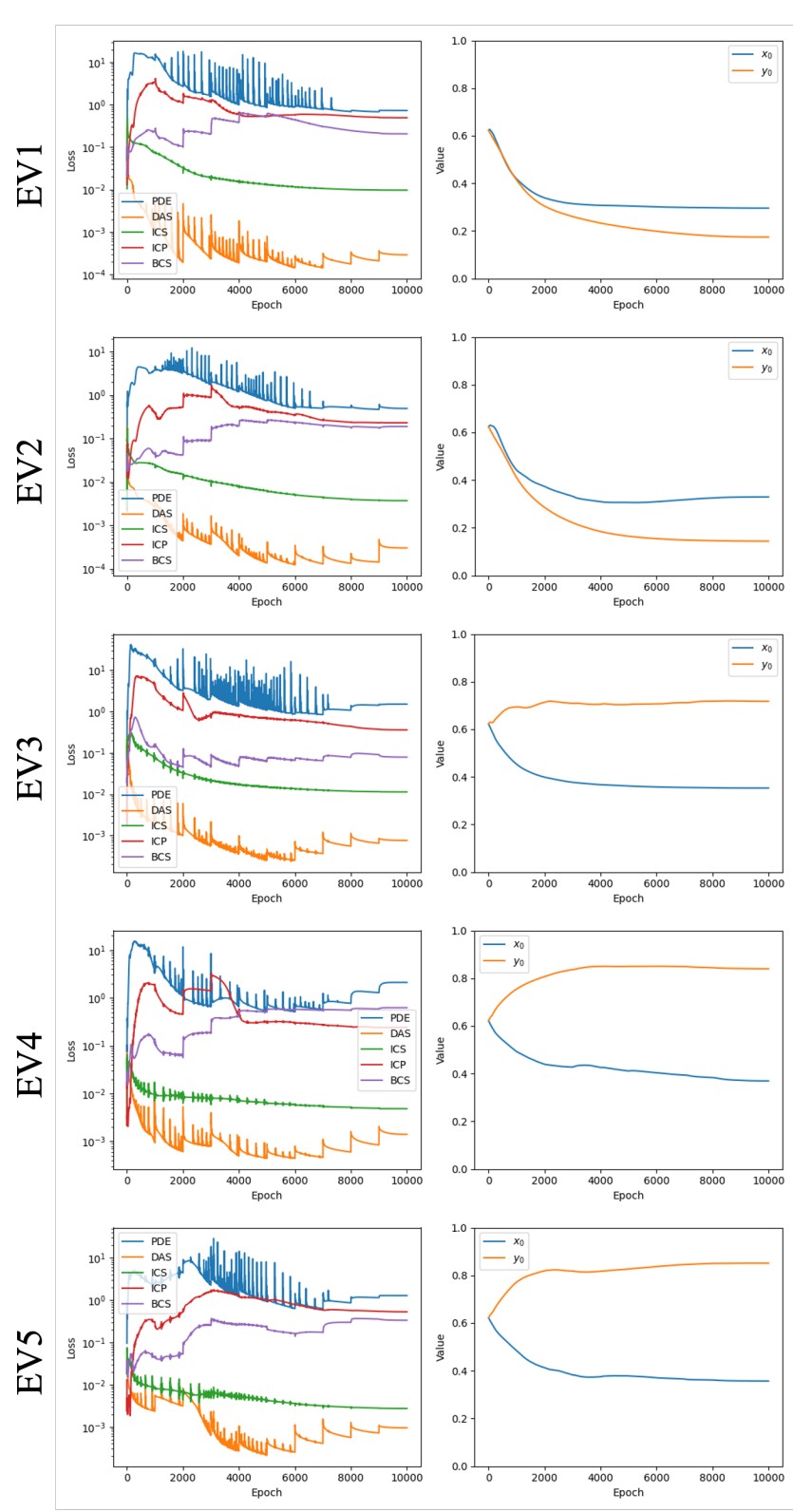

Figure 10: The left plot shows the convergence of individual loss components over epochs, including PDE loss, IC1(ICS) loss, IC2(ICP) loss, BC loss, and DAS loss. The right plot shows the convergence of the estimated source location coordinates $(x_0, y_0)$ over epochs.

