# OpenReview forum: "Integrating Distributed Acoustic Sensing and PINN Frameworks for Enhanced Indoor Sound Source Localization"
_ICLR.cc/2025/Conference — Submitted to ICLR 2025_

### Official Review · Reviewer_PcJK · 2024-10-23

**Soundness:** 2
**Presentation:** 2
**Contribution:** 1
**Rating:** 3
**Confidence:** 3

**Summary:**

The paper introduces the combination of distributed acoustic sensing (DAS) and physics-informed neural networks (PINNs). In DAS, observation stations are installed along standard optical fibers. These observation stations capture the anomalous propagation of light along the fibers. The anomaly source can both be internal (e.g., imperfections in the manufacturing process) and external (e.g., acoustic vibrations from the environment) and causes (undesired) backscattering of light. The authors use this deviation from normal light flow for localization of an unknown sound source. They use PINNs and integrate the acoustic wave equation alongside initial and boundary conditions. In model training, the network only uses acoustic pressure over time as input data, no ground-truth location is required. The authors test the combination of DAS+PINNs on synthetic data captured in 2D and 3D space. Qualitative results indicate the feasibility of their approach.

**Strengths:**

Combination of DAS and PINNs well motivated.

Two strategies of localization (incremental and full-data localization) seem very sensible.

In training for sound source localization, the proposed method does not require labels, possibly making it more widely applicable.

**Weaknesses:**

Weaknesses:
In my view, the paper has several weaknesses.

First, the paper only uses qualitative evaluation: Judging from the visualizations only (e.g., Fig4c), I cannot appropriately judge how well the approach works. There seems to be multiple possible sound sources identified by the algorithm, so speaking of "stable localization" or "accurately infer[ring] the sound location" seems unwarranted. Similarly, without a comparison to other methods, I cannot judge the performance of the DAS+PINN approach. To alleviate this, I suggest: 1) report MSE between predicted and actual sound source place, and 2) evaluate baseline methods: beamforming, MLP without the PINN part, and if possible also other ML approaches (see e.g. this paper: "Grumiaux et al: A SURVEY OF SOUND SOURCE LOCALIZATION WITH DEEP LEARNING METHODS")

Second, the presentation is lacking: there seems to be a missing paragraph after line 182, the description of the figures are too short, details about the training setup are missing, and mathematical symbols (lines 183ff) are not introduced properly (e.g., N_bc, N_ic, N_pde, x_pde). To alleviate this, I suggest: 1) add standalone figure descriptions on how to interpret them, 2) properly introduce mathematical symbols, 3) explain the network architecture of the PINN in writing, and 4) add a section on the training setup. Please also see my questions and suggestions for improvements further below.


Third, regarding the data, the connection between setting up the fiber cables (section 2, lines 124ff) and the synthetic tests (section 4) is unclear: are the test-spaces part of the overall installment? Or specifically installed? This example is only one of the unclearities that I have after reading the paper, please see my questions below. To alleviate this, I suggest: 1) a clarification on how the installment of the fiber cables relate to the synthetic tests, and 2) describing if the evaluated scenarios are synthetic, real, or idealized ones.


Fourth, the paper does not have paragraphs that I consider essential for (understanding) the paper: no treatment of the background of PINNs, a very short section on related work only, and no ethics statement (this is probably most critical, as the paper repeatedly stresses surveillance without compromising privacy. Alleviate this by: 1) adding a section on PINNs, 2) expanding the related work (on PINNs, but even more so on DAS), and 3) adding an ethics statement.


Fifth, the reference section of the paper includes only 13 references, which do not include other works on DAS (e.g., [1,2,3] found via a quick Google Scholar Search) and PINNs (e.g., [4,5,6]). I suggest you include these references, or others of your choice (this naturally will happen if my previous point is addressed).


Finally, it fails to convince me of the novelty: it appears to describe a trivial combination of DAS and PINNs, and also lacks ablation studies to asses the combinations effects. To alleviate this weakness, I suggest: 1) add a background on DAS, 2) add a background on PINNS, 3) describe the challenges (you had) in combining DAS and PINNs, and 4) discuss how the combination is superior in some way in applications.

**Questions:**

Questions:

In my understanding, one needs to place devices along the fiber cables to record the (anomalous) signals. Is that correct? This would probably be a disadvantage, as existing fiber layouts would need to be prepared beforehand and cannot be used as-is.

What are the "1250 distinct monitoring channels" (line 132f)? Are these the total installed recording stations?

Where can I see the channel distribution in Figure 1 (c.f. line 136)? Is each straight red line a channel?

What are the green squares in Fig4?

What is the "image source model" in line 95?

In your work, you repeatedly mention that DAS(+PINN) can be used for surveillance without compromising privacy. How does that work? If we know person X is in a room, we can locate and survey X with the proposed method? How does that relate to maintaining privacy?

Though the qualitative evaluation provided in the paper suggest that the method works, I am not sure how close to the real-world the test-cases are. Only test small rooms (up to 1m x 1m x 1m) are used the Halton-placement seems impractical: the room would then be clogged by fibers and not usable by, e.g., humans.

Figure 1: What is the "interrogator" marked in the figure and how does it relate to the paper?

Suggestions for improvement:

1. General suggestions

Adding a section on the background of PINNS would greatly improve the structure of the paper. Likewise, by expanding the related work on DAS, the paper would be better to comprehend.

Further, I would greatly appreciate a much expanded section on the training setup (epochs, optimizer, batch size, learning rate etc.). In similar spirit, explaining your data and its shape in more detail would improve my understanding of the setup.

For your figures, you could expand the description so they are comprehensible without having read the paper.

2. Specific suggestions
- lines 40ff: add further references to PINNs
- lines 43ff: consider adding more references to the application of PINNS or physical equations into machine learning in general, e.g. [7], [8]
- line 47: BCs and ICs used but not introduced beforehand --> consider introducing the abbreviation beforehand
- lines 48ff: consider adding references for the statement "Even when the loss function is minimized, the BCs and ICs are not strictly satisfied"
- line 49: consider a better transition between the two sentences; for me they do not seem to be related
- lines 57ff (challenges of PINNs): they do not seem to be related to the contributions (lines 75ff)
- lines 178ff: consider using a specific symbol for the neural network instead of the abbreviation NN.
- line 185: consider making the ordering of the equation parts consistent with the ordering of lines 167ff
- lines 270ff: Strategy 1 and Strategy 2 are mixed-up?
- lines 231ff: t_k versus just k in the exponent of the loss
- line 287 (and other places): abbreviation of FEM (for finite elements?) used without prior introduction --> consider introducing the abbreviation beforehand
- line 265: consider adding a short explanation about why the Halton-layout should be used (versus real random placement)
- Figure 2: Consider indicating the direction of data/information flow along the blue lines
- Figure 5: improve readability
- Figures 6, 7: consider improving the placement of the figures, possibly by shrinking Figure 7

Prior work on DAS:

[1]: Cedilnik, Gregor, et al. "Ultra-long reach fiber distributed acoustic sensing for power cable monitoring." Proceedings of the JICABLE. Vol. 19. 2019.

[2]: Jousset, Philippe, et al. "Fibre optic distributed acoustic sensing of volcanic events." Nature communications 13.1 (2022): 1753.

[3]: Lior, Itzhak, et al. "On the detection capabilities of underwater distributed acoustic sensing." Journal of Geophysical Research: Solid Earth 126.3 (2021): e2020JB020925.

Other works on PINNs:

[4]: Krishnapriyan, Aditi, et al. "Characterizing possible failure modes in physics-informed neural networks." Advances in neural information processing systems 34 (2021): 26548-26560.

[5]: Cuomo, Salvatore, et al. "Scientific machine learning through physics–informed neural networks: Where we are and what’s next." Journal of Scientific Computing 92.3 (2022): 88.

[6]: Raissi, M., Perdikaris, P., Karniadakis, G.E.: Physics-informed neural networks: A deep learning framework for solving forward and inverse problems involving nonlinear partial differential equations. J. Comput. Phys. 378, 686–707 (2019)

Suggested references for the introduction (lines 43ff)

[7]: Salvatore Cuomo, Vincenzo Schiano Di Cola, Fabio Giampaolo, Gianluigi Rozza, Maziar Raissi, and Francesco Piccialli. Scientific machine learning through physics–informed neural networks: Where we are and what’s next. Journal of Scientific Computing, 92(3):88, 2022.

[8]: Yogesh Verma, Markus Heinonen, Vikas Garg. ClimODE: Climate and Weather Forecasting with Physics-informed Neural ODEs. ICLR 2024. 2024

---

> ### Comment · Reviewer_PcJK · 2024-12-03
> **Response to request for commentary**
>
> Authors have partially responded to three (of four) previous reviews. I appreciate their efforts and have also seen that the paper has been expanded and improved upon in the revisions.
>
> However, for me the paper still lacks critical details, mostly related to the training setup, and a background on PINNs, plus the layout is lacking (arrangements of figures). I am therefore keeping my initial score.

---

> > ### Author Response · Authors · 2024-12-03
> > **Reply to Reviewer PcJK -- PINN background and training setup added.**
> >
> > We thank the reviewer for their continued feedback and for acknowledging the expansions and improvements we have made to the paper in the revisions.
> >
> > We understand that you still have concerns regarding the inclusion of critical details, mostly related to the training setup and a background on Physics-Informed Neural Networks (PINNs), as well as the layout and arrangement of figures.
> >
> > As per your request, we actually have added a detailed background on PINNs in the first paragraph of the paper and included the references you mentioned.
> >
> > We have also included an extensive description of the training setup in the Appendix. The section outlines the training procedures, and other relevant details necessary for reproducing our results.

---

> ### Author Response · Authors · 2024-12-03
> **we are still working on the reply to your comments**
>
> We have an official reply to all your questions on the way and will submit that soon. Sorry for the rush, please wait for a few hours until we get the official reply done. Thank you!

---

> ### Author Response · Authors · 2024-12-03
> **Reply to reviewer PcJK: Questions**
>
> We appreciate Reviewer **PcJK** for their questions that help us understand the contexts we haven't explained clearly. We use `Q#` to list the questions from Reviewer **PcJK**, and `A#` to list our responses.
>
> ### Questions
>
> 1. **Q1:** In my understanding, one needs to place devices along the fiber cables to record the (anomalous) signals...
>
>    **A1:** No, the devices need not be placed into fiber cables. The DAS method requires only a single laser source for both the transmission and reception of signals, and it does not necessitate additional devices along the fiber cable. The system works by continuously generating high-frequency pulse signals on one side of the optical fiber. As the signals propagate through the fiber, they produce reflected signals, and disturbances outside the fiber can affect the phase of these reflected signals. The DAS system can determine the location of disturbances on the fiber based on changes in phase. This means that the entire length of the fiber acts as a sensing element, allowing for the detection of micro-vibration signals over a large sensing area in real-time. This feature of DAS makes it highly adaptable and does not require the preparation or alteration of existing fiber layouts.
>
> 2. **Q2:** What are the "1250 distinct monitoring channels"...
>
>    **A2:** Yes, the 1250 channels are distributed along the fiber where we can obtain corresponding DAS data.
>
> 3. **Q3:** Where can I see the channel distribution in Figure 1 (c.f. line 136)? Is each straight red line a channel?
>
>    **A3:** No, the straight red lines represent merely the position of the optical fiber. The channels are uniformly distributed along the fiber.
>
> 4. **Q4:** What are the green squares in Fig4?
>
>    **A4:** The green squares indicate the converged locations under different initial guesses.
>
> 5. **Q5:** What is the "image source model" in line 95?
>
>    **A5:** The "image source model" is a method to account for the effects of reflections on the sound field. The basic idea of the image source model is to model the reflections of a sound source as if they were coming from additional, virtual sources, known as "image sources".
>
> 6. **Q6:** In your work, you repeatedly mention that DAS(+PINN) can be used for surveillance without compromising privacy...
>
>    **A6:**  The ``privacy-preserving'' means that the optical fibers can indicate human activities, e.g., moving trajectories of humans, but without intelligently understanding human speech due to the band limits. I have already added the explanation in the introduction section.
>
> 7. **Q7:** Though the qualitative evaluation ...
>
>    **A7:** Exactly, the halton-deployment is used to show the impacts of fiber geometry on localization. For real cases, it is more applicable to deploy fiber underneath the floor. In that case, the fiber can be distributed in 2D halton.
>
> 8. **Q8:** Figure 1: What is the "interrogator"...
>
>    **A8:** The term "interrogator" refers to the device that sends and receives signals in the DAS system.
>
> ---
>
> The replies to suggestions are in other replies due to the character limits.

---

> ### Author Response · Authors · 2024-12-03
> **Reply to Reviewer PcJK: Suggestions**
>
> Reply to **Suggestions**:
>
> ### Suggestions for Improvement
>
> We appreciate Reviewer **PcJK** for their thorough suggestions that help improve the original submission significantly. We use `S#` to list the suggestions from Reviewer **PcJK**, and `R#` to list our revisions in the manuscript.
>
> **S1:** Adding a section on the background of PINNs...
>
>    **R1:** We have added a detailed background section on PINN and related DAS work in the introduction.
>
>  **S2:** Further, I would greatly appreciate a much expanded section on the training setup...
>
>    **R2:** We have included an extensive description of the training setup in the Appendix. This section outlines the training procedures necessary for reproducing our results.
>
>  **S3:** For your figures, you could expand the descriptions so they are comprehensible without having read the paper.
>
>    **R3:** We have updated all figure captions and descriptions to be more detailed and self-explanatory.
>
>  **S4:** Lines 40ff: Add further references to PINNs.
>
>    **R4:** We have added more references to PINNs.
>
>  **S5:** Lines 43ff: Consider adding more references...
>
>    **R5:** Added.
>
>  **S6:** Line 47: BCs and ICs used but not introduced beforehand – consider introducing the abbreviation beforehand.
>
>    **R6:** Modified in Introduction.
>
>  **S7:** Lines 48ff: Consider adding references for the statement "Even when the loss function is minimized, the BCs and ICs are not strictly satisfied."
>
>    **R7:** Added.
>
>  **S8:** Line 49: Consider a better transition between the two sentences; for me, they do not seem to be related.
>
>    **R8:** We have revised the text to improve the flow.
>
>  **S9:** Lines 57ff (challenges of PINNs): They do not seem to be related to the contributions (lines 75ff).
>
>    **R9:** We have clarified the relationship in text.
>
>  **S10:** Lines 178ff: Consider using a specific symbol for the neural network instead of the abbreviation NN.
>
>  **R10:** We have introduced a specific symbol for the neural network in the equations for clarity.
>
>  **S11:** Line 185: Consider making the ordering of the equation parts consistent with the ordering of lines 167ff.
>
> **R11:** We have adjusted the ordering.
>
>  **S12:** Lines 270ff: Strategy 1 and Strategy 2 are mixed up?
>
> **R12:** We have corrected any inconsistencies in the naming of Strategy 1 and Strategy 2.
>
>  **S13:** Lines 231ff: \( t_k \) versus just \( k \) in the exponent of the loss.
>
> **R13:** We have standardized the notation to use \( t_k \) consistently.
>
> **S14:** Line 287 (and other places): Abbreviation of FEM (for finite elements?) used without prior introduction – consider introducing the abbreviation beforehand.
>
> **R14:** We have introduced the abbreviation FEM (Finite Element Method) when it first appears in the text.
>
> **S15:** Line 265: Consider adding a short explanation about why the Halton-layout should be used (versus real random placement).
>
>  **R15:** Added.
>
>    **S16:** Figure 2: Consider indicating the direction of data/information flow along the blue lines.
>
> **R16:** We have replotted Fig 2.
>
>  **S17:** Figure 5: Improve readability.
>
> **R17:** Adjusted.
>
> **S18:** Figures 6, 7: Consider improving the placement of the figures...
>
> **R18:** Fig 6 and 7 are combined. The new Fig 7 shows the real data results.

---

> ### Author Response · Authors · 2024-12-03
> **Reply to Reviewer PcJK: Weaknesses addressed**
>
> We thank Reviewer **PcJK** for their constructive comments that helps improve the paper a lot.
>
> ---
>
> ### Major Issue 1: Quantitative Performance Evaluation
>
> *Reviewer Comment:*
>
> *First, the paper only uses qualitative evaluation...
>
> **Response:**
>
> We appreciate the reviewer's suggestions regarding quantitative evaluation and baseline comparisons. Our main contribution lies in the **simultaneous localization of sound sources and reconstruction of the full spatiotemporal wavefield** using the DAS+PINN framework. Traditional methods such as beamforming, MLPs without physics-informed components, and other deep learning approaches mentioned typically focus solely on source localization without the capability for comprehensive wavefield reconstruction.
>
> Nonetheless, we acknowledge the importance of quantitative evaluation and have updated our manuscript to include location parameter convergence curves during training processes for both synthetic and real data cases. This provides a quantitative measure of our localization accuracy
>
> ---
>
> ### Major Issue 2: Paper Presentation
>
> *Reviewer Comment:*
>
> *Second, the presentation is lacking...
>
> **Response:**
>
> We have significantly improved the presentation of the paper:
>
> 1. **Figure Descriptions:** All figure captions have been expanded to be standalone, providing detailed explanations on how to interpret them without needing to refer to the main text.
>
> 2. **Mathematical Symbols:** We have thoroughly reviewed the manuscript to ensure all mathematical symbols (e.g., \( N_{\text{bc}}, N_{\text{ic}}, N_{\text{pde}}, x_{\text{pde}} \)) are properly introduced and defined upon their first occurrence.
>
> 3. **PINN Architecture Explanation:** We have added a detailed explanation of the network architecture of the PINN.
>
> 4. **Training Setup Section:** A comprehensive section on the training setup has been added in the Appendix.
>
> ---
>
> ### Major Issue 3: Real Fiber Setup
>
> *Reviewer Comment:*
>
> *Third, regarding the data, the connection between setting up the fiber cables (section 2, lines 124ff) and the synthetic tests (section 4) is unclear...
>
> **Response:**
>
> We have clarified the relationship between the fiber cable setup and our experiments:
>
> In Section 2, we have elaborated on how the fiber cables are installed and how they correspond to the real-world scenarios (new Section 5). The synthetic tests are idealized versions of actual installations.
>
> ---
>
> ### Major Issue 4: Background of PINNs
>
> *Reviewer Comment:*
>
> *Fourth, the paper does not have paragraphs that I consider essential for (understanding) the paper...
>
> **Response:**
>
> We have addressed your concerns by making the following revisions:
>
> 1. **Background on PINNs and Expanded Related Work:** We have added a comprehensive background on Physics-Informed Neural Networks (PINNs) and expanded the related work section in the introduction of the paper.
>
> 2. **Ethics Statement:**  We discuss how our method is privacy-preserving because, unlike microphones and cameras, DAS technology operates within limited frequency bands that do not capture intelligible speech or identify specific individuals. The method detects general acoustic events, such as movement trajectories, without recording personal conversations or identifying exact persons.
>
> By including these sections, we believe the paper now contains the essential elements for understanding our work and addresses the ethical considerations associated with surveillance technologies.
>
>
> ---
>
> ### Major Issue 5: References
>
> *Reviewer Comment:*
>
> *Fifth, the reference section of the paper includes only 13 references...
>
> **Response:**
>
> We have added numerous references to the paper, including key works on DAS and PINNs.
>
> ---
>
> ### Major Issue 6: Novelty Clarity
>
> *Reviewer Comment:*
>
> *Finally, it fails to convince me of the novelty...
>
> **Response:**
>
> We have addressed the concerns regarding the novelty of our work:
>
> 1. **Background on DAS and PINNs:** We have added detailed backgrounds on both DAS and PINNs in the introduction, setting the stage for our contribution.
>
> 2. **Challenges in Combining DAS and PINNs:** The main challenges include using sparse real DAS data to recover high-dimensional wavefields and simultaneously estimating unknown parameters like source locations. To address these, we developed two training strategies:
>
>    - **Strategy 1:** Focuses on accurate and fast source localization using early DAS data snapshots.
>    - **Strategy 2:** Aims for high-fidelity wavefield reconstruction by utilizing more DAS data over longer training times.
>
> 3. **Advantages of the Combination:** Our method enables simultaneous source localization and wavefield reconstruction without the need for labeled data, offering superior performance and efficiency in practical applications compared to existing approaches.
>
> ---
> The replies to **Questions** and **Suggestions** are in another two replies due to the character limits.

---

> > ### Comment · Reviewer_PcJK · 2024-12-03
> >
> > The authors have responded to all my questions and open points, and also have re-worked the paper in substantial parts. These changes have increased the quality of their paper (both in presentation of the results as well in the actual results). I appreciate their efforts.
> >
> > I might have been unclear in this regard, but I'd (still) appreciate a dedicated section on PINNs (not in the introduction, though thank you for adding this part). Further, the training setup is, in the current revision, not appearing in the appendix (maybe it has not been updated yet?).
> >
> > Taking the detailed responses and the authors' efforts into account, I am raising my score.

---

> ### Author Response · Authors · 2024-12-03
> **Reply to the Reviewer PcJK: PINN section and training setup issues**
>
> Thank you for your positive feedback and for acknowledging the efforts we've made to improve the paper. We're glad that the revisions have enhanced both the presentation and the results of our work.
>
> We apologize for any miscommunication regarding the dedicated section on PINNs. We understand now that you would prefer a standalone section rather than just an addition to the introduction. If given the opportunity to prepare a camera-ready submission, we will include a comprehensive section on PINNs to provide deeper insights into their role and implementation in our work.
>
> Regarding the training setup details, we acknowledge that Appendix A.2 currently only includes the convergence curves for individual losses and location parameters. Due to time constraints during the revision, we were unable to include the full details of the training setup. We assure you that in the final version, we will provide a detailed description of the training procedures, including hyperparameters, learning rate, optimizer, and other relevant information to enhance the reproducibility of our results. We appreciate your constructive feedback, which has been instrumental in improving our paper.
>
> Thank you for raising your score based on our revisions, and we hope to continue refining our work.

---

### Official Review · Reviewer_ddag · 2024-11-03

**Soundness:** 2
**Presentation:** 1
**Contribution:** 2
**Rating:** 3
**Confidence:** 3

**Summary:**

The paper suggests a method for sound source localization using measurements taken from deployed optic fibers inside building structures. The method is based on a physics-informed neural network that is quite small and simple, and does not require labeled data. Thus it has the potential to be used extensively. The authors provide the architecture of the network, but not very clearly. In addition, the method is only studied in computer simulations (not real measurements) and the experiments are limited to a single stationary source.

**Strengths:**

* Efficiency - suggesting a method for sound source localization by using already deployed optic fibers
* Simplicity - suggesting a method for sound source localization that does not require a lot of labeled data

**Weaknesses:**

* All experiments are based on a single static source, but moving target(s) is very common. It is not clear if this method is applicable to multi-source or moving sources. While showing results just for a single static source is OK, I would expect some explanation whether the method can be used for multi-source or to moving sources.
* The clarity of PINN architecture could be improved - while it is presented in Figure 2, I suggest explaining the inputs, outputs, and learned parameters explicitly.
  * Currently I find it difficult to understand how (x,y,z) is used as an input if we do not know the source position.. Are these initialized randomly according to some Gaussian distribution? Please make it clearer
  * In equation (11) we minimize over x_0, but this parameter is not clearly defined in the loss function, instead we have x_pde, x_bc, x_ic, x_data - what is the relation between these parameters?
* The presentation of both setup and results of the synthetic tests (section 4) could be improved
* Conclusions from experiments
  * Lacking a clearly defined evaluation criteria (source localization error)
  * Authors mention the importance of optimal design of DAS array - (a) they did not address optimal source placement, (b) is controlling the placement really possible? Isn’t this given by the structure of the optical fiber array (which is placed according to other constraints)?
  * It is mentioned that training strategy 1 significantly outperforms strategy 2 but this is not clear from Figure 4. Performance looks very similar. Why did the authors conclude otherwise? Please explain. In addition, if there is such a difference, why is this the case? Can we use this insight to understand when is better to use strategy 1 over 2?
  * Section 4.2 is not presented clearly. In addition, there are many conclusions with no results shown
    * Line 355 - “Strategy 2 converges with the correct 2D source location…” - can you please provide numerical errors?
    * Lines 357-359 - where do we see the performance of each strategy?
* While the paper claims it deployed an impressive fiber network (figure 1) - this is not used anywhere in the experiments. All experiments are based on computer simulations of optic fibers behavior.

**Minor comments**
* Line 47 - BC and IC abbreviations are not defined (boundary and initial conditions?)
* Line 50 - XPINNs and FBPINNs are defined only later
* Line 199 - p_{data} is defined before equation (10), this is somewhat confusing. Is this the pressure term at the last line of equation (7)?i.e., “...+2p(x,t)E||22”?
* Figure 2 - the two plots below the loss equations are two small and unclear. If they provide some clarity, please incorporate them in a clearer way
* I would suggest to present equation (8) before equation (7) - I think it clearer to understand the strain rate along the axial direction of the fiber before this is used in the los function of the network
* Figure 3
  * Mention what is the the red Gaussian-like distribution (source location)
  * The right-hand side of both plots (a) and (b) are not defined - what do we see there?
* Figure 4 - please describe what we see in the plots, i.e., what are the green boxed, red star, and the difference between the right- and left- hand side plots
* Figure 5 - the colors are very bright and it is difficult to understand the visualization of sound propagation. In addition, the pressure colorbar is missing units.
* Lines 270-271 - I think that you have mistakenly mixed between the two strategy indices, i.e., it should be “the Incremental Time-Stepping Training Strategy (Strategy 2) significantly outperforms the Simultaneous Training with All Data Strategy (Strategy 1) in achieving stable localization”

**Questions:**

* What are the trade-offs in performance between strategy-1 and strategy-2 training?
* How is training performed when there is a moving source? How fast can the network learn and produce the correct source positions? What about multiple sources?
* I left additional questions regarding the experiments under “weaknesses” section

---

> ### Author Response · Authors · 2024-12-03
> **Reply to Reviewer ddag: Real moving target experiment added**
>
> We thank Reviewer **ddag** for their constructive comments and appreciation of our strengths.
>
> ### Major Issue 1: Moving Target(s)
>
> In the latest version of our article, we include experiments with moving targets. In the experiment, a person walks within a square room. We tracked the person's movement using the proposed method.
>
> For multi-source problems, our PINN framework assumes a Gaussian initial source time function. This means that when multiple sources happen very close to each other, the inversion will return one event location -- a surrogate single source. If multiple sources happen separately, the problems can be treated as single-source localization problems.
>
> ---
>
> ### Major Issue 2: PINN Clarity
>
> In the latest version of our article, we have provided a detailed explanation of the PINN architecture, including inputs, outputs, learnable parameters, and the specific form of the loss function. We have tested convergence results under various possible initial guesses for the source position, as shown in Figure 4. In equation (11), \( p_{\theta} \) actually refers to \( p_{\theta;x_0} \). Here, \( x_0 \) is a learnable parameter in the source time function of the initial conditions, which is part of the loss of the initial conditions. We have clarified all the parameters in the revised version.
>
> ---
>
> ### Major Issue 3: The Presentation of Synthetic Tests
>
> In our latest article, we have added more detailed explanations regarding the synthetic tests.
>
> ---
>
> ### Major Issue 4: Conclusions from Experiments
>
> **Q1:** Lacking a clearly defined evaluation criteria (source localization error).
>
> **A1:** We have shown the convergence curves of source locations for both synthetic and real data.
>
> **Q2:** Authors mention the importance of optimal design of DAS array - (a) they did not address optimal source placement, (b) is controlling the placement really possible?
>
> **A2:** Our synthetic tests show the impacts of array geometry on location accuracy. Controlling fiber location is possible depending on the engineering objectives, then we can optimize fiber deployment based information gains.
>
> **Q3:** It is mentioned that training strategy 1 significantly outperforms strategy 2 but this is not clear from Figure 4...
>
> **A3:** It can be seen in Figure 4 and 6 that Strategy 2 performs better converging to the true solution compared to Strategy 1.  Strategy 2 uses a time-stepping approach for training, and since the distribution of sound waves is more concentrated at earlier times, it can better capture the source information compared to Strategy 1.
>
> **Q4:** Section 4.2 is not presented clearly. In addition, there are many conclusions with no results shown.
>
> **A4:** We have added more detailed explanations in Section 4.
>
> **Q5:** Line 355 - "Strategy 2 converges with the correct 2D source location..." - can you please provide numerical errors?
>
> **A5:** In the inversion problem we are concerned with, the differences between these two strategies can be intuitively compared from Figure 4, and the numerical errors are not as critical.
>
> **Q6:** Lines 357-359 - where do we see the performance of each strategy?
>
> **A6:** We have added more detailed explanations in Figure 4 to discuss the different results between the strategies.
>
> **Q7:** While the paper claims it deployed an impressive fiber network (Figure 1) - this is not used anywhere in the experiments. All experiments are based on computer simulations of optic fibers behavior.
>
> **A7:** We have included real-world experimental data in our latest article.
>
> ---
>
> ### Minor Issues
> We have addressed all these minor issues in the revised version.
>
> ### Questions
>
> **Q1:** What are the trade-offs in performance between strategy-1 and strategy-2 training?
>
> **A1:** When accurate and fast source localization is the priority, Strategy 1 is generally better, achieving precise localization results using only a few early DAS data snapshots; however, it may not capture the full details of the wavefield. In contrast, when detailed wavefield reconstruction is needed, Strategy 2 obtains a high-fidelity reconstruction by utilizing more DAS data snapshots over a longer training time, but it may not localize the source as quickly or may converge to mirror source locations.
>
> **Q2:** How is training performed when there is a moving source?
>
> **A2:** In our latest article, we demonstrate the effectiveness of our method using real DAS measurements.
>
> **Q3:** How fast can the network learn and produce the correct source positions?
>
> **A3:** We have added the training process in our latest article. Normally with only a few thousand steps, our DAS PINN can converge. Every 1000 epochs take ~46 seconds with one A800 GPU.
>
> **Q4:** What about multiple sources?
>
> **A4:** This has been answered before.
> ---
>
> We appreciate the reviewer's feedback, and we hope that the revisions satisfactorily address the concerns raised.
>
> If there are any further questions or suggestions, we would be happy to address them.
>
> ---

---

### Official Review · Reviewer_RFvc · 2024-11-03

**Soundness:** 3
**Presentation:** 3
**Contribution:** 2
**Rating:** 5
**Confidence:** 2

**Summary:**

The paper presents a source localization method based on distributed acoustic sensing using optical fibers. They propose to use physicals-informed neural networks with a loss function that is based on the sound wave propagation and the boundary conditions. Two strategies for real-time localization are presented, using all the data at once or in an incremental manner.
Simulation results demonstrate the capabilities of the proposed method.

**Strengths:**

The paper deals with the emerging field of distributed acoustic sensing through optical fibers, and its use for source localization. Harnessing physics-based deep models is a promising approach. The simulations in 2D and 3D highlight the potential of the proposed method for solving the source localization problem.

**Weaknesses:**

Overall, the innovation of the paper is more application specific and less in terms of developing a new ML method (which can be specific or general). The use of PINN for sound field reconstruction was already proposed in Olivieri et al. (2024). They state that the main difference is that here they add impedance boundary conditions, which seems like a small addition in terms of innovation, as far as I understand.
The paper only contains results on simulated data, while the abstract claims: “Using real indoor DAS measurements, we demonstrate the effectiveness of our approach”, and in the intro: “We demonstrate the simultaneous localization of sound sources and recovery of the spatiotemporal acoustic field using *real* DAS measurements”. Additionally, it is unclear why they describe fiber placement (shown in Fig 1.), while they did not apply their method on this data.
In addition, the results contain no comparison to baselines and no quantitative results.

**Questions:**

Several questions related to the weaknesses mentioned above:
1.	Is the key contribution of the paper in terms of methodology or the specific application?
2.	How do they deal with the challenges mentioned in the intro (lines 57-64)?
3.	Why aren't there any real-world experiments as stated at the beginning of the paper?
4.	Consider adding comparisons to baselines.
5.	Consider adding quantitative results in terms of localization error.
Additional minor questions:
1.	How are the weights in Eq. (7) chosen?
2.	How does impedance and boundary conditions are determined?
3.	Are Halton and Volume configurations realistic as they require fibers covering the entire space?
4.     Fig. 3  - in the caption, what is the meaning of the acronym FEM?
5.     Fig. 4.  - in the caption, what is presented in the left plot in each subplot?
6.     Figs. 4 and and 7, why are there several final locations?
7.	Consider adding an ablation study comparing to optimization without boundary conditions.

---

> ### Author Response · Authors · 2024-11-27
> **Reply to Reviewer RFvc: Key contribution issue and the application of the method to real DAS data**
>
> We thank Reviewer **RFvc** for their constructive comments and appreciation of our strengths.
> ### Weaknesses
> We acknowledge the reviewer's concern regarding the novelty of our work. While Olivieri et al. (2024) introduced the use of Physics-Informed Neural Networks (PINNs) for sound field reconstruction, our work differentiates itself in several significant ways:
>
> - **Integration with Distributed Acoustic Sensing (DAS)**
> - **Simultaneous Localization and Reconstruction Using Real Data**
> - **Novel Training Strategies**
>
> Regarding the impedance boundary conditions, while they are an essential component for accurately modeling wave reflections in enclosed spaces, they are not the sole novelty of our work. We included impedance boundary conditions to capture the reflective properties of the room boundaries, which is critical for realistic wavefield reconstruction.
>
> We apologize for the confusion caused by the initial submission. Due to time constraints, we were unable to include the real data analysis in the original manuscript. We have since updated the paper to address this concern comprehensively:
>
> Added Real Data Analysis: We have included a new section titled "Real Data" where we apply our DAS-PINN framework to real DAS recordings. The results demonstrate that our method can successfully track the indoor human footsteps, validating the practical applicability of our approach.
>
> ---
>
> ### Questions
>
>  **Q1: Is the key contribution of the paper in terms of methodology or the specific application?**
>
>    **A1:** The key contribution of our paper lies in both the development of a novel methodology and its practical application. Methodologically, we introduce a physics-informed neural network (PINN) framework integrated with Distributed Acoustic Sensing (DAS) technology, enabling simultaneous localization of sound sources and reconstruction of spatiotemporal acoustic fields without the need for labeled data. From an application perspective, we demonstrate the effectiveness of our method using real DAS measurements, showcasing its potential for advanced indoor sensing applications such as real-time tracking of moving targets.
>
> **Q2: How does impedance and boundary conditions are determined?**
>
>    **A2:** We did not determine the impedance of the wall through measurements. Instead, we assumed a constant impedance coefficient \( Z = 7462 \) Pa·s/m to represent typical walls.
>
> **Q3: Fig. 3 - in the caption, what is the meaning of the acronym FEM?**
>
>    **A3:** FEM stands for "Finite Element Method". We apologize for not explaining this acronym earlier. We have modified the caption of Figure 3 to include the full term.
>
> **Q4: Fig. 4 - in the caption, what is presented in the left plot in each subplot?**
>
>    **A4:** In Figure 4, the **left plot** in each subplot presents the localization results during the training process. The black circles show the initial guess for the source location. The green squares represent the final estimated source locations after the last training epoch, while the red star marks the true source location. The curves connecting the black circle and the green square show the trajectories of the estimated source locations as they converge toward the final solution during training. This plot illustrates how the initial guesses evolve over training epochs and demonstrates the effectiveness of the PINN framework in accurately localizing the sound source.
>
> **Q5: Figs. 4 and 7, why are there several final locations?**
>
>    **A5:** The multiple final locations depicted by the green squares in Figures 4 and 7 correspond to the outcomes of the localization process starting from different initial guesses for the source location. By showing multiple final locations, we highlight the importance of appropriate initializations and training strategies to guide the model toward the true source location.
>
> **Q6: Consider adding an ablation study comparing to optimization without boundary conditions.**
>
>    **A6:** We appreciate the suggestion to include an ablation study comparing optimization with and without boundary conditions. In our experiments, optimization without boundary conditions is effectively represented during the early training epochs of the *Incremental Time-Stepping Training Strategy* (Strategy 2), where we use only the first few time snapshots. At these early time steps, the acoustic waves have not yet propagated far enough to interact with the walls, so the boundary conditions have minimal impact on the wave propagation. This scenario mimics an environment without boundary reflections. We have added a paragraph at the end of Section 3 to highlight this connection.
>
> ---
>
> We are grateful for the reviewer's feedback, which has helped us improve the clarity and completeness of our manuscript. We hope that the revisions satisfactorily address the concerns raised.
>
> If there are any further questions or suggestions, we would be happy to address them.

---

> > ### Comment · Reviewer_RFvc · 2024-12-02
> >
> > Thank you for your response.
> > Since part of my concerns were answered (adding experiments on real data and clarifying the contribution), I raise my score.

---

### Official Review · Reviewer_gyvL · 2024-11-03

**Soundness:** 3
**Presentation:** 3
**Contribution:** 2
**Rating:** 5
**Confidence:** 3

**Summary:**

This paper presents a physics-inspired neural network for estimating the location of an acoustical source using the deformations of optical fibres as input. This is an innovative approach, and I have not seen similar approaches before. The physical basis is presented clearly and with appropriate detail. The text and grammar are mostly good, but variables, acronyms and figures are not thoroughly described. The balance between thorough physical modelling and brief experiments feels awkward.

**Strengths:**

- Contributions of the article are very clearly specified
- The connection between physics and neural networks in the application scenario is a strong contribution

**Weaknesses:**

Major issues:
First, the correspondence between the real-world location of fibre optic cables and the Halton distribution is weak at best. Second, we probably do not have the accurate location of a fibre optic cable. Yet both the distribution and accurate location of measurement points are required for accurate results. Thus, the correspondence between the experiment and real-world scenarios is weak. More realistic scenarios would have helped.
- This paper claims to be about using fibre optic cables as microphones, but it contains only experiments with piece-wise linear and random arrays. The only connection to fibre optics is the underlying physics model. Experiments closer to fibre optics scenarios would help a lot.
- The balance of the paper is skewed since the physics side is strong, but experiments are brief.

Minor issues:
- All text, and especially equations in Figure 2, are very small.
- Interpretation of symbols in Fig 3-4 are not explained.

**Questions:**

All my questions have been resolved in the revision.

---

> ### Author Response · Authors · 2024-11-27
> **Reply to Reviewer gyvL: Real data section added**
>
> We thank Reviewer **gyvL** for their constructive comments and appreciation of our strengths, such as:
>
> - **"This is an innovative approach, and I have not seen similar approaches before. The physical basis is presented clearly and with appropriate detail."**
> - **"Contributions of the article are very clearly specified."**
> - **"The connection between physics and neural networks in the application scenario is a strong contribution."**
>
> ---
>
> ## Response to Major Issues
>
> ### **Major Issue 1: Realistic Scenario**
>
> > *The correspondence between the real-world location of fiber optic cables and the Halton distribution is weak at best. Second, we probably do not have the accurate location of a fiber optic cable. Yet both the distribution and accurate location of measurement points are required for accurate results. Thus, the correspondence between the experiment and real-world scenarios is weak. More realistic scenarios would have helped.*
>
> You are absolutely correct that the experiments shown in the original version are less realistic. The Halton distribution is just an extreme case in which the DAS array is randomly distributed in space, imagining a highly cluttered room filled with fiber-optic cables.
>
> For accurate locations of indoor fiber optic cables, we can obtain the locations using the documented fiber deployment plan and **tap experiments** conducted before data acquisition. In tap experiments, we use a hammer to tap specific locations near the fiber, and the DAS signal monitor shows a strong pulse at the corresponding channel along the cable. These channels are spaced along the fiber at fixed spatial intervals.
>
> We have incorporated a realistic scenario using real DAS data collected in rooms with floor plans shown in **Figure 1**. Subplots of **Figure 1** now include real data corresponding to a walking person. Additionally, a new section has been added to the revised version to present results using this real acquired data.
>
> ---
>
> ### **Major Issue 2: Experimental Data**
>
> > *This paper claims to be about using fiber optic cables as microphones, but it contains only experiments with piece-wise linear and random arrays. The only connection to fiber optics is the underlying physics model. Experiments closer to fiber optics scenarios would help a lot.*
>
> We sincerely appreciate the reviewer's insightful feedback regarding the need for experiments that closely mimic real-world fiber optic scenarios. To address this, we have added a dedicated section presenting experiments using real DAS data collected from an actual fiber optic deployment.
>
> This section demonstrates the practical application of our framework in realistic settings and complements the strong physics foundation of the paper with detailed real-world experiments.
>
> ---
>
> ## Response to Minor Issues
>
> ### **Issue 1: Small Text in Figures**
>
> > *All text, and especially equations in Figure 2, are very small.*
>
> We have reproduced **Figure 2** with larger text for improved readability.
>
> ---
>
> ### **Issue 2: Missing Symbol Explanations**
>
> > *Interpretation of symbols in Figures 3-4 is not explained.*
>
> We have updated **Figures 3-4** with detailed explanations of all symbols. The newly added figures also include these updates.
>
> ---
>
> ## Response to Questions
>
>  **Q1:** *In Sections 1 and 2, it is unclear why sensing acoustics with optical fibers is privacy-preserving. It would be useful to mention how using optical fibers as cheap microphones is better for privacy than using proper microphones.*
>
> **A1:** Apologies for the confusion. By "privacy-preserving," we mean that optical fibers can detect human activities, such as movement trajectories, without being able to intelligently interpret human speech due to the limited frequency band. This advantage is now explicitly explained in the introduction section.
>
> ---
>
>  **Q2:** *In the introduction, what are BCs and ICs? Please explain acronyms before using them.*
>
> **A2:** BCs and ICs stand for **boundary conditions** and **initial conditions**, respectively. We have added these definitions in the introduction for clarity.
>
> ---
>
>  **Q3:** *In Equation 7, what are all the different variables? Please explain, especially the meaning of subscripts.*
>
> **A3:** Thank you for pointing this out. We have revised the manuscript to include a detailed explanation of all variables and subscripts in **Equation 7**, ensuring their roles in the loss function are clear and comprehensible.
>
> We hope the revised manuscript addresses all your concerns and suggestions. Thank you again for your valuable feedback!

---

### Meta-Review · Area_Chair_HfSj · 2024-12-22

**Metareview:**

The paper presents the use of physics-informed neural networks (PINNs) for solving the governing equations for distributed acoustic sensing (DAS) with initial and boundary conditions. Experiments are presented on real and synthetic data for the task of localizing acoustic sources from the DAS data and demonstrate promising results.

The paper received four reviews, all inclined towards reject. While the reviewers appreciated the innovation in applying PINNs on DAS data,  the reviews spelled out several glaring issues in the paper (summarized below). AC agrees with the issues pointed out by the reviewers, especially on the novelty of the approach, which while is innovative from a practical standpoint, appears to be a direct application of PINNs for a new task without any significant contribution or insights furnished from a machine learning perspective. As such, AC recommends reject.

**Additional Comments On Reviewer Discussion:**

Below is a summary of the discussion that ensued, the changes the authors made to the paper, and outstanding issues that remain unresolved.
1. The approach is an application of PINNs for a new task, rather than deriving any new machine learning approach (RFvc, PcJK)
2. Missing evaluation criteria, qualitative nature of evaluation, missing comparisons to baseline methods (RFvc, ddag, PcJK, gyvL)
3. Lack of clarity in technical details and experimental setup (ddag, gyvL)

The paper was thoroughly revised catering to the reviewers' questions, with inclusion of additional experiments in the appendix, improved details on the evaluation criteria, and the experimental setup. Authors also added real world experiments and new results using moving targets. However, baseline comparisons were not provided and the nature of evaluations/performance improvements remain more on the qualitative side.

---

### Decision · Program_Chairs · 2025-01-22

Reject